# How Are the Links between Alcohol Consumption and Breast Cancer Portrayed in Australian Newspapers?: A Paired Thematic and Framing Media Analysis

**DOI:** 10.3390/ijerph18147657

**Published:** 2021-07-19

**Authors:** Amy Rudge, Kristen Foley, Belinda Lunnay, Emma R. Miller, Samantha Batchelor, Paul R. Ward

**Affiliations:** Discipline of Public Health, Flinders University, Adelaide 5001, Australia; amy.r.rudge@gmail.com (A.R.); Kristen.foley@flinders.edu.au (K.F.); Belinda.lunnay@flinders.edu.au (B.L.); Emma.miller@flinders.edu.au (E.R.M.); stur0028@flinders.edu.au (S.B.)

**Keywords:** breast cancer, alcohol, media analysis, Australian newspapers, framing analysis

## Abstract

A dose-dependent relationship between alcohol consumption and increased breast cancer risk is well established, even at low levels of consumption. Australian women in midlife (45–64 years) are at highest lifetime risk for developing breast cancer but demonstrate low awareness of this link. We explore women’s exposure to messages about alcohol and breast cancer in Australian print media in the period 2002–2018. Methods: Paired thematic and framing analyses were undertaken of Australian print media from three time-defined subsamples: 2002–2004, 2009–2011, and 2016–2018. Results: Five key themes arose from the thematic framing analysis: Ascribing Blame, Individual Responsibility, Cultural Entrenchment, False Equilibrium, and Recognition of Population Impact. The framing analysis showed that the alcohol–breast cancer link was predominantly framed as a behavioural concern, neglecting medical and societal frames. Discussion: We explore the representations of the alcohol and breast cancer risk relationship. We found their portrayal to be conflicting and unbalanced at times and tended to emphasise individual choice and responsibility in modifying health behaviours. We argue that key stakeholders including government, public health, and media should accept shared responsibility for increasing awareness of the alcohol–breast cancer link and invite media advocates to assist with brokering correct public health information.

## 1. Introduction

Despite clear epidemiological evidence that alcohol consumption is a risk factor for breast cancer [1], women’s awareness of the alcohol–breast cancer link remains considerably low [2]. While portrayals of the alcohol–breast cancer link within the media could be an avenue to increase women’s awareness, there has been conflicting messages as to whether alcohol is a harmful or helpful substance to health generally. Media-disseminated health information can affect the public’s behavioural response to reducing individual risk factors [3]. However, conflicting and changing information about health risks in the media adversely impacts public perception and trust [4]. This paper analyses newspaper reporting of the links between breast cancer and alcohol to examine how messages that women are receiving are framed in the media. This is crucial since, for example, if the link between alcohol and breast cancer is framed as ‘certain’, ‘unknown’ or ‘unlikely’, then that will influence how women respond. Additionally, if the link is framed as the responsibility of the government, the alcohol industry, individual women or open to ‘fate’, this will impact ‘what’ women might do to reduce their risk of breast cancer.

Against this backdrop, we know that targeting a modifiable risk factor such as alcohol is a good approach to reducing the disease burden of breast cancer [5,6]. Baade and colleagues [7] estimate that approximately 40,000 cancer cases and 40% of breast cancer cases in 2025 in Australia could be prevented if improvements were made to increasing awareness of modifiable risk factors. Understanding awareness levels regarding alcohol as a modifiable risk for breast cancer and developing appropriate policy and prevention strategies is therefore a critical public health priority. The challenges of addressing this issue among midlife women (aged 45–64 years), those most at risk of developing breast cancer, may be significant given recent research that articulates the clear role that alcohol plays in the lives of women within this demographic [8].

Alcohol has been described as an ‘integral part’ [9] of the Australian ‘national identity’ [10], occupying an iconic place in Australian culture, with far-reaching health consequences [10]. Alcohol commonly features in social gatherings, serving as a means to celebrate and commiserate [9]. As a legal, socially acceptable, and widely accessible psychoactive substance, the Australian government employs a harm minimisation approach to regulate consumption [9,10]. To this end, the National Health and Medical Research Council (NHMRC) established guidelines to reduce alcohol-related health risks [11]. The guidelines aim to enable the public to make informed decisions regarding alcohol consumption, by educating them about the relationship between alcohol intake and short-and long-term risks.

Despite the NHMRC guidelines, alcohol consumption in Australia is high, relative to other high-income countries [12]. The most recent National Drug Strategy Household Survey (NDSHS) [13] shows females in their 50s were the most likely of all age groups to drink at levels that increased lifetime risk of harm. The survey findings also revealed that, regardless of their consumption levels, many Australians thought of themselves as an “occasional, light or social drinker” (p. 47), believing alcohol consumption that exceeded the Australian guidelines would not pose a health risk. Meyer et al. [2] showed that middle-aged (45–64 years) women’s awareness of the alcohol–breast cancer link is low and women consume alcohol for various purposes, many of which were viewed by the women as health enhancing [8,14]. Adults in another study perceived cancer as ‘inevitable’, seeing little or no reason to change consumption behaviours despite the knowledge that ‘alcohol causes cancer’ [15].

Concomitantly, the alcohol industry promotes the notion of alcohol as a ‘normal, everyday product’ and women specifically are targeted by alcohol marketing, obvious through the feminisation and ‘pinking’ of alcohol products [16]. In Australia, alcohol industry group ‘Drinkwise’ campaigns identify Australian culture [17,18] as responsible for fuelling the ‘drinking problem’, simultaneously drawing attention away from commercial interests which impact on, and profit from, women’s alcohol consumption [19]. The alcohol industry plays on corporate social responsibility initiatives to raise awareness of the importance of ‘responsible drinking’ [18], rather than modifying or reducing consumption. Evidence suggests that the media is an ineffective way to communicate the risk relationship between alcohol and cancer in the past [2], potentially contributing to women’s limited awareness of the alcohol–breast cancer link.

The purpose of this research was to explore whether/where/how the relationship between alcohol and breast cancer has appeared in Australian newspapers to identify recommendations of how the alcohol–breast cancer link can be more effectively portrayed in the media, with the intent of raising awareness for women of this link, rather than contributing to further confusion.

## 2. Materials and Methods

A media analysis was undertaken using an innovative analytical approach developed by Foley et al. [20] designed specifically for public health media inquiry. Whilst there are numerous content analyses of print and social media, our analysis is innovative due to its paired qualitative thematic and framing analysis. The method combines thematic and framing analyses, enabling more in-depth insight into how alcohol was communicated in the media as a modifiable breast cancer risk factor. This compound analysis is critically important because it encourages viewing the topic from different perspectives and examines how messages were framed [21], which has substantial implications for how women might then respond [20,22]. Identifying themes in the content then tracking changes over time using the (qualitative) framing analysis to determine ‘how’ the content is framed by the media will help elucidate how women may have viewed messages regarding the alcohol–breast cancer link.

### 2.1. Sampling

Print newspapers were selected as the source for analysis. While newspaper readership has declined over previous years, the proportion of Australians who continue to regularly read newspapers has remained stable [23]. More than half of all Australian adults (76.9%) in the period 2018–2019 accessed newspaper material across available platforms (printed, websites, social media and Apps) [24]. Research also shows when accessing health information through the media, sources which require ‘active’ consumption (e.g., newspapers and magazines) are more likely to act as primary health sources (i.e., people are unlikely to go to the research that underpins the media stories) [25].

When identifying relevant time periods, we utilised contextual information to signpost where points of interest about alcohol–breast cancer messaging might be observed in the data. The initial time period identified was 2009–2011—a ‘peak’ period due to the release of three significant public health documents which brought attention to alcohol harms. These included updated NHMRC drinking guidelines (2009) which reflected new understandings of the health impacts resulting from even low levels of alcohol consumption [26]; an updated monograph from the International Agency for Research on Cancer (IARC) on alcohol consumption, identifying breast cancer as causally related [27], and the publication of a position statement on alcohol as a carcinogen from The Cancer Council Australia, officially recommending Australians abstain from alcohol in order to reduce cancer risk [28]. We hypothesised newspaper reporting on the release of these documents would mean Australians might have been exposed to newspaper messaging regarding low risk levels of consumption, the carcinogenicity of alcohol (with a particular focus on breast cancer), and that there is no safe level of alcohol consumption regarding cancer risk.

Two time points either side of this ‘peak’ were then identified as comparative time periods. A ‘pre’ period, 2002–2004, four to seven years prior to the release of the documents was chosen. In the period 2002–2004, the IARC classified alcoholic beverages as a Group 1 carcinogen [27] but had yet to identify evidence to include breast cancer as an alcohol-related cancer [27,29]. This period provided a ‘control’ with minimal content on the alcohol–breast cancer link expected to appear in newspapers. Next, the period 2016–2018, defined as ‘post’, was selected to determine how newspapers represented information about the alcohol–breast cancer link four to seven years after the dissemination of key public health documents.

The newspapers with the highest readership, representing the two major corporations, were included, along with their Sunday paper counterparts (Figure 1). The Factiva database was used to search for newspaper articles in the three time periods, using the keywords ‘*alcohol*’, ‘*beer*’, ‘*wine*’, ‘*spirits*’ or ‘*drinking*’ in conjunction with ‘*cancer*’ or ‘*tumour*’, as well as ‘*breast*’. A total of 1047 newspaper articles were identified within the three time periods. Details of all newspaper articles were entered into a spreadsheet and assessed for eligibility, searching for articles directly discussing the consumption of alcohol in relation to breast cancer. Over 800 articles mentioned the key search terms separately and did not discuss alcohol in relation to breast cancer (e.g., recipes with chicken breast and wine). These articles, in addition to duplicates Factiva had missed, were excluded. The remaining 203 articles commented on alcohol and cancer, of which 153 directly referred to breast cancer in relation to alcohol consumption specifically. Thus, 153 newspaper articles were included for analysis (2002–2004 *n* = 50; 2009–2011 *n* = 66; 2016–2018 *n* = 37).

### 2.2. Analysis

Thematic and framing analyses were combined following Foley et al.’s methodology [20,30]. Firstly, thematic analysis was undertaken, which is an inductive ‘ground-up’ approach, whereby the researcher commences organisation of the data from a ‘blank slate’ endeavouring to interpret and draw meaning from the data [31]. Using NVivo (version 11, QSR International Pty Ltd., Melbourne, Australia), the articles were separated according to the three time periods and examined separately, in chronological order. Open coding of the content of each newspaper article per time frame revealed broad categories and hundreds of codes. Axial coding explored further within these to classify the open codes into categories. Finally, selective coding looked across the axial categories to characterise the patterns observed and emerging themes. During the latter two stages of analysis, initial insights and emerging patterns were documented in matrices, then refined in conjunction with authors K.F., E.R.M. and P.R.W. These preliminary interpretations aided in shaping the analytical process.

Secondly, framing analysis facilitated exploration of how stories were presented and shaped in the newspapers, through the identification of included or excluded information, to subsequently influence audience understanding and interpretation [32]. Drawing on Entman’s framing definitions, we considered four components to framing (Figure 2). Framing analysis makes visible the interaction of these framing components and frames within the media. Foley et al. [20] articulates the benefits of coding data fragments rather than whole texts, in order to develop their own innovative approach to framing. This adaptation reported herein focused primarily on coding data fragments within news articles rather than the whole article, in order to capture the complex layers of framing present in whole articles. This is completed manually to allow for the identification of subtle meaning within the text, followed by a process of deductively coding data fragments to the coding framework developed a priori. The frames identified a priori were: ‘medical’, ‘behavioural’ and ‘societal’—proposed by Foley et al. [20] as ‘integral to public health’ (p. 5). Previous research also features variations of these frames (e.g., biomedical, lifestyle, social) as common categories and approaches within the public health field [33,34,35]. During the framing analysis, each article was re-examined and data fragments coded by hand in NVivo to the most salient frame component, as defined by Entman [36], within the relevant frame. As with the thematic analysis, an initial interpretation of the framing analysis was documented in tables and reviewed with authors KF, EM and PW for reliability. A table was drafted for each frame to present the most salient frame components within that frame and four differing shades were used to indicate the prominence (no prominence, mild prominence, moderate prominence, strong prominence) of each frame component in the frame across the three time periods.

## 3. Results

Across the time periods of analysis, several changes were noted in the portrayal of the alcohol–breast cancer link. This was demonstrated through the number of articles published in each time period, the emphasis on varying themes that emerged from the thematic analysis, as well as how news articles were positioned within medical, behavioural or societal frames across these time periods.

As anticipated, many of the articles which discussed alcohol consumption and breast cancer were published during the peak period 2009–2011 (*n* = 66), compared to the pre-period 2002–2004 (*n* = 50) and post-period 2016–2018 (*n* = 37) (Figure 3). Surprisingly, the proportion of articles which mentioned the alcohol–breast cancer link was not higher in the post-period than the pre-period (Figure 3). The article content addressing the alcohol–breast cancer link between 2016 and 2018 did more closely reflect the key public health documents.

### 3.1. Results of the Thematic Analysis

The thematic analysis revealed five key themes present to varying degrees across the three time periods. The most prominent themes—Individual Responsibility and False Equilibrium—were strongly positioned in each time frame. Meanwhile, the themes Ascribing Blame, Cultural Entrenchment, and Recognition of Population Impact were more subtly portrayed during the pre-period (2002–2004) but became more apparent during the peak (2009–2011) and post- (2016–2018) periods. An important finding was that in in the period 2002–2004, the benefits of alcohol to health were quite prominently featured, whereas in in the period 2016–2018, any such benefits were portrayed as ‘not worth the potential risk of developing breast cancer’.

#### 3.1.1. Ascribing Blame

Over the years, newspapers portrayed the relationship between alcohol consumption and breast cancer as inherently an individual ‘female’ problem. Breast cancer, resulting from alcohol consumption was identified as an outcome associated with being of the female sex. The choices females made and the interactions of these decisions (for example, to bear children, or breastfeed) with their unique biological design were ‘blamed’ for increased breast cancer incidences.


*She said drinking had contributed to rising numbers of breast cancer cases, although other factors such as the trend for women to remain childless or have smaller numbers of children and not breastfeed had probably had a bigger effect*
(Daily Telegraph, 14 November 2002).

The impact of hormone replacement therapy (HRT) in the alcohol–breast cancer risk mix was often reported during this time:


*But women whose mothers or sisters had breast cancer, or those taking post-menopausal oestrogen replacement, are at greater risk from alcohol*
(Herald Sun, 30 August 2004).

By 2016–2018, the focus shifted from the agency of older women and those of child-bearing age to young females prior to their first pregnancy: 


*Young women who drink alcohol before their first pregnancy face a 35 per cent higher risk of developing breast cancer, new research suggests*
(Daily Telegraph, and Herald Sun, 21 October 2016).

Consistent across time periods was the focus on biology and the interaction of alcohol with female hormones (oestrogen) as well as a focus on post-menopausal females who were considered at higher risk of breast cancer from alcohol consumption: 


*Other risks for post-menopausal women were: obesity after menopause, as estrogen after menopause is produced in fat cells, alcohol consumption, some risk for those that were non-identical twins and some risk for those with a family history of breast cancer*
(Sunday Herald Sun, 6 April 2003).


*Experts aren’t clear on why alcohol increases the risk of breast cancer. One theory is that alcohol increases the levels of oestrogen in the blood, which is a risk factor for developing breast cancer*
(The Australian, 9 June 2017).

By presenting females as actively making choices related to childbearing, breastfeeding and hormone treatment, these articles tended to isolate inherent female factors and ascribe a level of blame to females for the increase in alcohol-related breast cancer incidence.

#### 3.1.2. Individual Responsibility

Newspaper content from 2009 onwards changed focus from female biological mechanisms to addressing the risk of alcohol consumption behaviours with responsibility for adhering to cancer prevention recommendations positioned within individuals’ control. Although this theme was present across all three time periods, it was most prominent in the articles from 2009 onwards. Alcohol was portrayed as a ‘dangerous’ substance and women were warned to be wary of consuming it. However, the risk alcohol posed in relation to breast cancer was consistently described as modifiable, with individual choice and responsibility highlighted. Women were urged to make necessary changes to reduce their risk:


*Women should still remain more wary than men when it comes to drinking, however, and not just because of their smaller body size*
(Sunday Herald Sun, 16 November 2003).


*“You might not be able to help your genes but you can make lifestyle choices.”*
(Sydney Morning Herald, 2 May 2009).

Newspaper content that contained recommendations for breast cancer prevention similarly focused on modifying individual behaviours. However, the suggestions provided were often non-specific (e.g., ‘reduce your intake’), and conflicted with previous articles and/or the drinking guidelines at the time (e.g., ‘consume no more than one standard drink’ when guidelines and articles cite two standard drinks), or were vague, including non-standard units of alcohol measurement (e.g., ‘more than the equivalent of half a bottle of wine a week’). Therefore, the implications of different patterns and levels of alcohol consumption in relation to breast cancer risk were unclear:


*Heavy alcohol consumption is particularly dangerous, with women drinking more than a bottle of wine a day at 40 to 50 per cent higher risk of the disease*
(Daily Telegraph, 14 November 2003).


*Even women who had three to six drinks a week had a 15 per cent increased risk of breast cancer compared with non-drinkers*
(The Australian, 26 November 2011).


*This amount of alcohol was equivalent to three teaspoons of wine per day, she added*
(Herald Sun, 4 May 2017).

It was not until 2016–2018 that a slight shift toward population-level recommendations was observed in how the newspapers portrayed the alcohol–breast cancer link (described in the Recognition of Population Impact theme below). In the period 2016–2018, articles acknowledged that the public may be less familiar, if not ‘ignorant’ of the relationship between alcohol consumption and breast cancer:


*The findings are likely to come as a surprise to many Australians, who are well versed on the dangers of tobacco, but remain ignorant of alcohol’s link to mouth, throat, stomach, bowel, breast and liver cancer*
(Sydney Morning Herald, 14 July 2018).

Within newspaper content from each period, the individual was positioned as an informed decision maker about this relationship:


*“It’s about informing them so they can make informed choices.”*
(Daily Telegraph, 1 December 2004).


*“You should certainly be aware of the information available to make an informed decision and, if you do drink, do so in moderation.”*
(Daily Telegraph and Herald Sun, 4 February 2017).

##### Cultural Entrenchment

Newspaper discussion in all periods placed a level of accountability for the increase in alcohol-related breast cancers on cultural values which encouraged and normalised alcohol use. Modern and more affluent ‘lifestyles’ were specifically criticised in newspaper reports and blamed for the harm alcohol consumption contributes to increased breast cancer prevalence:


*Australia’s middle-class lifestyle could also be contributing to the incidence of some cancers such as breast cancer*
(The Australian, 15 December 2004).


*Modern lifestyles which feature regular drinking, lack of exercise and increased obesity are fueling the disease’s rise, the European Breast Cancer Conference heard*
(Daily Telegraph, 27 March 2010).

From 2009–2011 onwards, abstinence was portrayed as an unreasonable expectation due to how deeply entrenched alcohol is in women’s everyday lives:


*Alcohol is often part of everyday life, and it can be hard to avoid it completely*
(Sunday Telegraph and Sunday Herald Sun, 24 January 2010).


*It’s unrealistic to recommend to patients that they completely abstain from alcohol*
(The Australian, 9 June 2017).

Consequently, alcohol-related breast cancers were initially portrayed as attributable to a lifestyle which promotes consumption (2002–2004 and 2009–2011) but from 2009–2011 onwards, lifestyles featuring alcohol were presented as unavoidable considering cultural drivers to consume.

#### 3.1.3. False Equilibrium: Alcohol as Tonic and Poison

Alcohol was portrayed as both a ‘tonic’ and ‘poison’ in all time periods, possessing the dichotomous ability to both protect and harm health. Throughout the period 2002–2004, newspapers portrayed alcohol-related harms/benefits as a counterbalance. Whereby, ‘too much alcohol’ put the drinker at risk of breast cancer, while moderate consumption, particularly of good wine, was portrayed as a way to prevent breast cancer, and protect health:


*The evidence supports theories that a moderate daily intake of wine helps prevent stroke and heart disease, as well as perhaps diabetes and prostate and breast cancer*
(Daily Telegraph, 26 October 2002).


*All women can weigh the benefits of drinking alcohol against the slight increased risk of breast cancer*
(Sydney Morning Herald, 13 February 2003).

In the period 2002–2004, and to a lesser extent in the period 2009–2011, messages that alcohol consumption might pose risk for breast cancer was presented alongside content which incorrectly claimed the risk could be mitigated through diet modifications, or that any purported health benefits of alcohol outweighed the risk of breast cancer:


*The increased danger to individual women, however, is usually outweighed by the significant benefits to cardiovascular health*
(The Australian, 16 June 2004).


*Eating plenty of folate, a B vitamin found in spinach and broccoli, for example, may reduce the risk of breast cancer associated with alcohol*
(Sydney Morning Herald, 7 May 2011).

Interestingly, the risk of breast cancer from alcohol consumption was presented in the periods 2002–2004 and 2009–2011 as greater than some medical treatments and served to justify the use of HRT and supplements:


*“If you have no symptoms I wouldn’t take it, but the risk of HRT causing breast cancer is really minimal—and people often forget other risk factors such as drinking, or a high fat diet.”*
(The Australian, 4 September 2004).


*Although combined forms of HRT may slightly increase the chances of breast cancer, the effect is dwarfed by other risk factors, such as obesity, diet and alcohol*
(Sydney Morning Herald, 7 February 2009).

In the period 2009–2011, articles about the alcohol–breast cancer link included research findings which claimed the beneficial properties of wine extended beyond prevention to include breast cancer treatment:


*Red wine could help women with breast cancer boost the chances of their treatment being successful*
(Daily Telegraph, 16 February 2011).


*An ingredient in red wine can stop breast cancer cells growing and may combat resistant forms of the disease*
(Sunday Herald Sun, 2 October 2011).

While in the periods 2002–2004 and 2009–2011 the benefits of alcohol were portrayed to outweigh the risks of breast cancer if consumed responsibility, reports of alcohol as protective and beneficial to health faded over time. By 2016–2018, the release of key public health publications seemed to have impacted this balance, and alcohol consumption was no longer considered ‘worth the risk’ of breast cancer. In the period 2016–2018, there was only a single mention, and the article concluded that the risks associated with ‘moderate’ alcohol consumption outweighed any purported health benefits:


*Is a glass of red wine a day ok? No: This is one “wish list” rule everyone hopes is true. However, health professionals claim benefits in red wine are outnumbered by the negatives in alcohol as a whole. The “red wine is good for you” bandwagon began due to the fact it contains the natural compound resveratrol, which acts like an antioxidant and helps prevent damage to the blood vessels in your heart and reduce your LDL, or bad cholesterol. While this may be true, Australian Medical Association vice-president Dr Tony Bartone says research shows a link to cancer*
(Daily Telegraph and Herald Sun, 4 February 2017).

#### 3.1.4. Recognition of Population Impact

From 2009–2011 onwards, there was moderate recognition of the increasing population-level impact of alcohol consumption. Articles drew attention to the association of a ‘common cancer’—breast cancer—to alcohol consumption and noted the substantial influence that alcohol could therefore have on reducing the population-wide alcohol-related breast cancer disease burden:


*In Australia, 5000 or 5 per cent of cancers, including one in five breast cancers, are attributable to long-term, chronic drinking*
(The Age, 19 September 2011).


*Alcohol is estimated to cause more than 1500 cancer deaths in Australia every year, with breast and bowel cancers of particular concern… About 1330 bowel cancers and 830 breast cancers are attributed to alcohol each year in Australia, and it has been estimated about 3 per cent of cancers can be blamed on alcohol consumption each year*
(Sydney Morning Herald, 8 September 2017).

In addressing the population impact of alcohol on breast cancer incidence, a small number of newspaper articles presented population-level interventions as appropriate solutions:


*The research will help shape future campaigns warning of the risks of drinking. Graphic labelling similar to those displayed on cigarette packs could also be introduced to minimise harmful drinking*
(Herald Sun, 20 April 2017).


*It may be beneficial for public health authorities to consider guidelines specific for cancer survivors, rather than relying on prevention messaging*
(Herald Sun, 4 May 2017).

From the thematic analysis, five themes emerged which presented alcohol as simultaneously harmful and beneficial to breast cancer, as well as deeply embedded in Australian culture, and contributing to a considerable morbidity and mortality burden. Meanwhile, alcohol-related breast cancers were positioned as stemming from inherent female concerns and choices controlled by the individual which were perceived to be the responsibility of the individual to address. Additionally, various conflicting and misleading messages of the potential health harms/risks associated with alcohol consumption were portrayed by newspapers.

### 3.2. Results of the Framing Analysis

Using framing analysis of newspaper articles identified news fragments in terms of the extent to which they defined problems, diagnosed causes, made moral judgements, or suggested remedies under a medical, behavioural or societal frame (within each three time periods) [36]. The prominence of content situated within a medical frame waned after the period 2002–2004, while content coded at the societal frame was most salient in the period 2016–2018. Content that fit within the behavioural frame was the most prominent across the three time periods.

#### 3.2.1. Medical Framing

News fragments focusing on biological interactions of alcohol within the body and resulting pathophysiological effects [35] were grouped together into the medical frame. In this frame, alcohol and breast cancer were approached from an objective medical science perspective. This frame was most prominent in the period 2002–2004, when alcohol was considered more beneficial than harmful, and certain individual female characteristics were seen to increase alcohol’s breast cancer risk. Discussions of alcohol’s carcinogenic classification did not appear until 2009–2011. At this time, reports were beginning to balance the pros and cons of alcohol’s health impacts. By 2016–2018, the risk of breast cancer from alcohol was still used to justify HRT but mentions of the beneficial properties of alcohol were no longer present. Alcohol was now framed as a convincing carcinogen linked to breast cancer, with women post-menopausal or who had never given birth, deemed most at risk (Table 1). Four shades are included in the scale, from lighter (no prominence), through mild prominence and moderate prominence, to darker (strong prominence).

#### 3.2.2. Behavioural Framing

News fragments categorised to the behavioural frame addressed individual choices and behaviours [34,35]. Here, the frame components of causal attribution, moral evaluation and treatment recommendation took greater prominence, reflecting newspaper portrayals featuring alcohol and breast cancer which centred around the influence women themselves had and could have going forward. Over the three time periods, the newspaper portrayals of the alcohol–breast cancer link was most prominent within the behavioural frame, but the content differed over time. Initially, how much individuals drank was identified as the cause, and reductions in consumption subsequently recommended. At this stage, possible benefits of alcohol were discussed, but it was the responsibility of the individual to find balance between the harms and benefits. In the period 2009–2011, as the portrayal of perceived benefits began to wane, individual choices were further condemned, especially among breast cancer patients and survivors. These individuals were the focus of numerous articles which criticised their decisions and behaviours as dismissive of the alcohol–breast cancer warning by continuing to drink after receiving breast cancer diagnoses. As of 2016–2018, the focus remained firmly on individual behaviours. Here, an increased subset of women was portrayed as being at higher risk of breast cancer from alcohol consumption, and the decisions they could and should make in the future were of particular focus (Table 2).

#### 3.2.3. Societal Framing

News fragments categorised to the societal frame referenced community concerns of alcohol-related health harms and broad structural factors contributing to these harms [35,37]. In this instance, frame components were concerned with the consumption of alcohol in society and the impact this had on the population-level breast cancer incidence. In the earlier two periods, attention was drawn to lifestyles normalising regular alcohol consumption and highlighted their involvement in the increase in alcohol-related breast cancer cases. The continuing consequences of increasing consumption were also a focus in the period 2016–2018, where low public awareness of the alcohol–breast cancer link was acknowledged and cited as resulting from a lack of campaigns and health messages. Over time, the reported societal recommendations coincided with the publication of research reports describing interventions, which still relied on individuals to initiate behavioural changes (e.g., graphic labelling to raise awareness and reduce consumption). Throughout all time periods, however, alcohol’s perceived position as a ‘default setting’ in Australian society never faltered (Table 3).

## 4. Discussion

Our discussion focusses on how portrayals of the alcohol–breast cancer link change in newspapers over time and the potential implications that this has for women’s awareness and public health more generally. Accurately representing the relationship between alcohol consumption and increased breast cancer risk via the media can be considered a shared responsibility between the media, the government, the alcohol industry, and individuals [38,39], because it can improve public awareness of health risks and thus help people make informed decisions when consuming alcohol. Interpretation of our findings needs to be explored with this in mind. As a popular, accessible source of health information, newspapers ‘have the power to change public perceptions on health-related issues’ [40] p. 39. The media can exert multiple influences on individuals through: agenda setting and pointing to issues of public interest; the selection and salience of issues; circumscribing (even indirectly) community and individual attitudes to risk; and suggesting political action/s [41]. Some researchers assert [42,43,44,45,46] that contemporary public mistrust of scientists and their research is due to their findings being misrepresented by the media. Divergences between scientific evidence about health and its portrayal in the media contribute to a misleading media landscape and may also have implications for the beliefs and values of consumers [47].

Our findings are consistent with previous research examining media coverage of breast cancer and alcohol, whereby journalists were found to ‘cherry pick’ when reporting findings of scientific studies by paying particular attention to controversial findings of single studies, thereby contributing to conflicting messages in the print media [48]. Additionally, Eliott et al. [49] found, where an acknowledgement of the relationship between alcohol and cancer was presented in Australian newspapers from 2005 to 2013, it was often overshadowed by competing health claims. Within the 2002–2004 and 2016–2018 periods we analysed, there remained several articles that reported on the purported benefits of alcohol in the prevention and treatment for breast cancer, presented various proportions of risk of breast cancer resulting from alcohol consumption, and suggested differing levels of alcohol intake to reduce the risk of alcohol-related breast cancer. This is despite clear evidence that breast cancer was classified as causally related to alcohol consumption [27]; that no safe threshold of consumption was identified to avoid cancer [28]; and the national guidelines set the maximum limit for reducing long-term alcohol-related harm at two standard drinks per day [26].

The way the media ‘frames’ the link between alcohol and breast cancer is critical as it can shape future behaviours of the ‘responsible parties’ and define whose responsibilities they are. Our analyses showed that the behavioural frame took prominence, with women consistently portrayed as responsible for health decision making. The medical and societal framing of the alcohol–breast cancer link was generally neglected by the media, with structural factors infrequently ascribed as contributing to increased risk of breast cancer resulting from alcohol. The medical frame featured mildly in the period 2002–2004 but waned shortly thereafter, and the societal frame only began to emerge in the period 2016–2018. The privileging of the behavioural factors results in an unbalanced portrayal of the alcohol–breast cancer link and obscures some of the structural forces of disease risk as well as those medical issues beyond individual control. This can contribute to increased blame (and self-blame) for individuals that develop breast cancer, while also neglecting opportunities to raise and leverage public awareness and action on the commercial determinants of health. While it does appear that the societal angle was drawn on more by media in more recent times, integrating information that links health outcomes with population changes (i.e., exploring why women in midlife drink more than in previous decades) would be of vital assistance to contextualising alcohol consumption as a modifiable risk factor for breast cancer.

### 4.1. Implications of Our Study Findings

We focus this section particularly on the implications of our study for governments, the alcohol industry, and media organisations.

#### 4.1.1. Implications for Media Organisations

The media has a shared responsibility to inform the Australian population of alcohol-related breast cancer risks and prevention strategies. Due to the media’s reliance on academic research and the simultaneous reliance of researchers on the media to promote their findings, there is an opportunity for these parties to restructure the current media landscape and focus on health outcomes [40]. Dedicated public health advocates who can navigate the space between research and public dissemination and promote a more balanced media representation would be of great benefit [50,51,52]. Such advocacy would facilitate the clarity and consistency of health findings reported in the media [49]. Additionally, opportunities exist for public health professionals to respond to conflicting messages portrayed by the media in relation to the alcohol–breast cancer link. For example, through the implementation of mass media campaigns, the public health sector can promote community health messages to enhance the media’s portrayal of the relationship between alcohol and breast cancer—ensuring accurate and helpful content is disseminated, avoiding claims that represent the relationship in an ambiguous or contradictory manner [53]. Moreover, increasing awareness of this link might garner public support for other policy options.

#### 4.1.2. Implications for the Alcohol Industry

Strategies employed by the alcohol industry were evident in media stories analysed. Just as the newspaper articles reflected alcohol to be culturally entrenched and the responsibility of the individual to curb consumption and reduce health risks, so too does the alcohol industry promote alcohol as a part of everyday Australian society and target individual behaviours when disseminating ‘drink responsibly’ messages [17,18,54]. Directing attention away from organisational responsibility (and the severity of an issue), and towards personal responsibility, has been identified as a key tactic employed when ‘unhealthy’ products (such as alcohol) are involved [55,56]. This is observed within the alcohol industry, where DrinkWise frame’s alcohol-related harms as the result of cultural drinking practices [18]. In their position statement, the Australian Medical Association [38] proposed the alcohol industry take responsibility for the harms alcohol use contributes to, and not profit from ‘excess alcohol use’. The alcohol industry therefore needs to acknowledge the alcohol-related cancer harms and employ suitable strategies to raise awareness that every drink poses a health risk.

#### 4.1.3. Implications for Governments

Since we completed this analysis, the NHMRC drinking guidelines have been revised, now acknowledging ‘the less you drink, the lower your risk of harm from alcohol’ [11]. Whilst this demonstrates some level of government accountability for addressing alcohol-related harm [57], the impetus for change remains with the individual. Whilst education regarding alcohol-related harm is important, it does not necessarily lead to behavioural change. Our analysis found the media strongly emphasise the behavioural frame, aligning with the neoliberal political environment which positions individuals as responsible for health decision making, conversely relieving institutions of accountability [58]. Our findings suggest other policy levers should be utilised to address alcohol harm, beyond individual responsibility, including understanding and addressing population-level changes in alcohol consumption for women in midlife. The use of ambiguous terms such as ‘drink responsibility’ or ‘sensible drinking’ by the government in light of findings that there is no threshold for risk-free drinking [59] is vexed.

### 4.2. Study Limitations

This study only examined how messages were portrayed to the reader rather than how it is received—we cannot make assertions that suggest the content is interpreted by all women in the same way [36,60]. Building on this, the analysis requires the meaning to be derived from abstract variables and is reliant on the researcher’s interpretation, which might differ if undertaken by another researcher [61,62]. Our approach of cross-checking analysis and interpretations amongst co-authors aimed to generate collective insight on the findings. Additionally, the sample was limited to mainstream Australian print media. While this is representative of the cross-platform media distributed by Australian newspapers and the targeted age-group [23], news platforms are evolving and becoming increasingly varied, therefore attracting different sub-populations [63,64]. Other themes or frames may emerge when the same topics are presented on different media platforms to different audiences (e.g., social media). For example, a media analysis of social media platforms would enable insight into the cancer prevention content received by young adults, public responses to, and interactions with, such information, and potential platforms for future interventions [65,66]. It is possible that women’s magazines may also provide more extensive and trusted information to their readership [25]. However, there was no such database at the time of data collection. If Australian women’s magazine content is added to media databases in the future, a comparison of content could make a valuable contribution to this field. Although the purpose of this study was not to gauge the views of women about how the media portrays the alcohol–breast cancer link, future research could examine this to better understand the influence of the media on women’s understandings of knowledge about alcohol as a breast cancer risk factor. Such studies could also identify the most influential themes or frames in prompting behaviour change or facilitating support for interventions designed to reduce alcohol-related cancer harms [25].

## 5. Conclusions

Media can influence the beliefs and behaviours of their audience as well as direct public discussion and political objectives. How the media represents the relationship between alcohol consumption and breast cancer is therefore important for informing women of alcohol-related breast cancer risk. This study demonstrates that the Australian print media has not consistently or accurately portrayed the alcohol–breast cancer link, instead exposing readers to conflicting and controversial news stories focused predominately on individual behaviours. This study identified the most prominent themes and frames portrayed in Australian newspapers over three time periods, representative of the message disseminated by the media about the alcohol–breast cancer link. Over three time periods, five themes were revealed: Ascribing Blame, Individual Responsibility, Cultural Entrenchment, False Equilibrium, and Recognition of Population Impact. Further, one frame was most prominent across all time periods: Behavioural. The commonality between the findings of the thematic and framing analyses was the focus on the individual choices and behaviours which contributed to the alcohol-related breast cancer burden, and subsequently their responsibility to enact changes to reduce this burden. Furthermore, the representation of the alcohol–breast cancer link was found to be incomplete and contradictory—new articles commonly reported results or outdated claims which conflicted with evidence-based information about the causal relationship between alcohol and breast cancer. Through media advocacy and public health promotion, media platforms can be better utilised to disseminate more balanced representations of the relationship between alcohol and breast cancer as well as the wider range of avenues to address the structural drivers of this disease.

## Figures and Tables

**Figure 1 ijerph-18-07657-f001:**
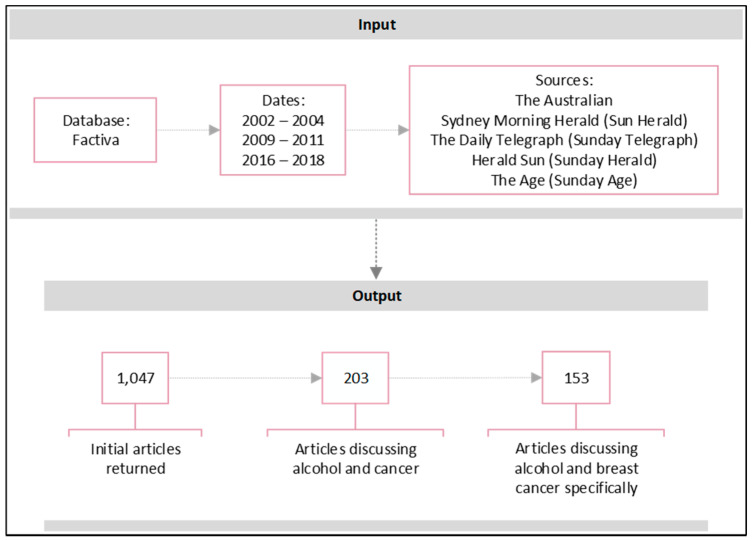
Input and output of Factiva database search.

**Figure 2 ijerph-18-07657-f002:**
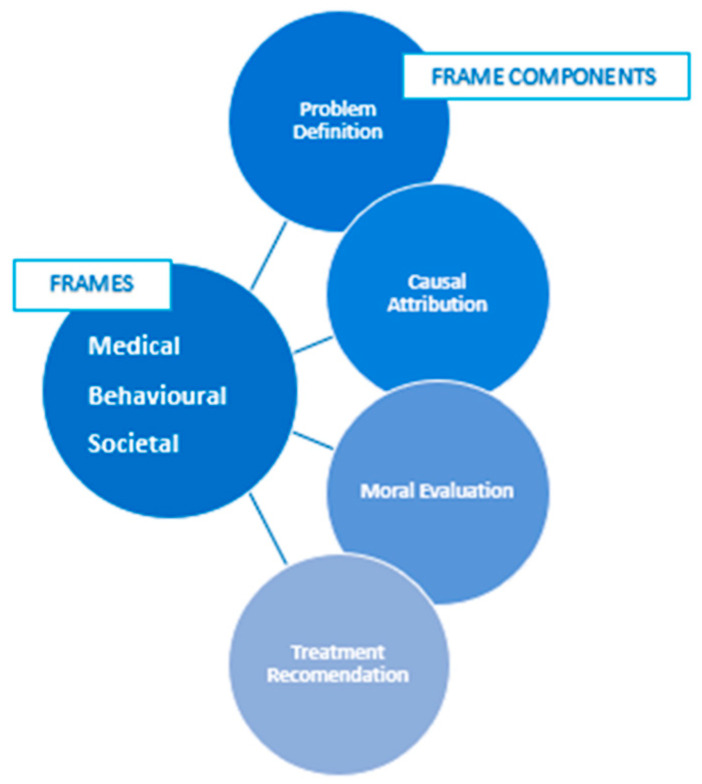
Frames and frame components included in framing analysis.

**Figure 3 ijerph-18-07657-f003:**
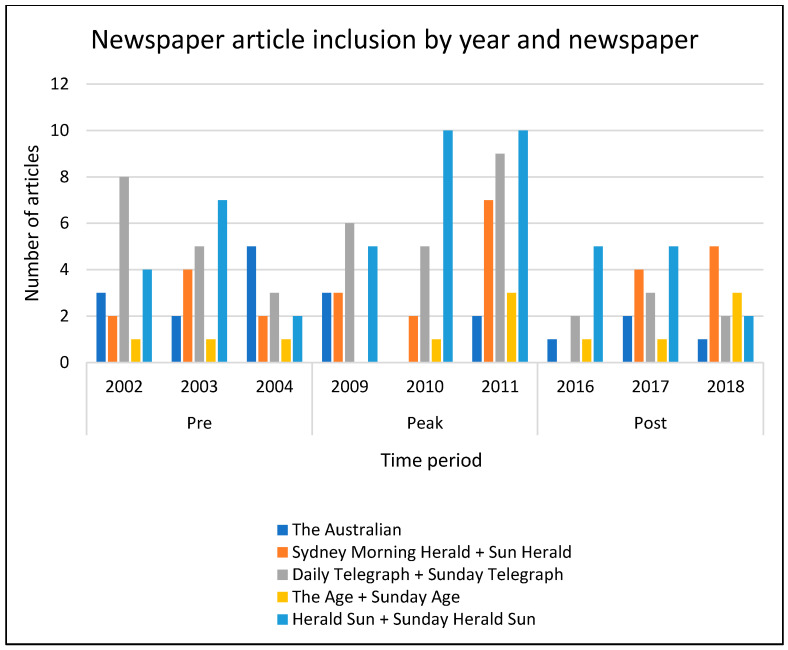
Newspaper articles included in media analysis by year and newspaper.

**Table 1 ijerph-18-07657-t001:** Dominant medical frame components.

Medical Frame				
	Problem Definition	Causal Attribution	Moral Evaluation	Treatment Recommendation
**2002–2004**	Breast cancer incidence increasing	Increases risk—older women, those taking HRT, and with family history most at risk Alters oestrogen levels		Good for health Risks can be mitigated Benefits outweigh risks HRT use justified Most at risk should avoid alcohol
**2009–2011**	Link recently identified	Carcinogenic Increase oestrogen levels and damage DNA	Confusion—simultaneously beneficial and harmful Treating a ‘life threatening ailment’	Treats cancer Use of HRT and supplements justified
**2016–2018**	Breast cancer morbidity and mortality increasing—alcohol attributed	Carcinogenic ‘Convincing’ evidence Post-menopausal, and nulliparous women most at risk Increase oestrogen levels and damage DNA		HRT use justified

Four shades are included in the scale (from light to dark): no prominence, mild prominence, moderate prominence, strong prominence.

**Table 2 ijerph-18-07657-t002:** Dominant behavioural frame components.

Behavioural Frame				
	Problem Definition	Causal Attribution	Moral Evaluation	Treatment Recommendation
**2002–2004**		Consumption increasing risk Lifestyle choice Risky drinking—higher levels	Women’s choices contributing to rise Should be wary There are consequences Need to weigh risks and benefits	Some consumption good for health Avoidance or reduction recommended Include alcohol-free days
**2009–2011**	Fuelling breast cancer incidence	Lifestyle choice Patient’s drinking habits Risky drinking—lower levels	Bad choices are a threat to health Risk can and should be avoided Need to worry more Women not deterred Women blamed	Some consumption good for health Make better choices Avoidance or reduction recommended Include alcohol-free days
**2016–2018**	Alcohol abuse increasing breast cancer incidence Rise in female alcohol-related breast cancer deaths	Consumption known risk factor Higher risk when drinking before first pregnancy Risky drinking—higher levels	Make informed decisions Risk can and should be avoided Always think about risk Patients making bad decisions	Avoidance or reduction recommended Avoidance best option

Four shades are included in the scale (from light to dark): no prominence, mild prominence, moderate prominence, strong prominence.

**Table 3 ijerph-18-07657-t003:** Dominant societal frame components.

Societal Frame				
	Problem Definition	Causal Attribution	Moral Evaluation	Treatment Recommendation
**2002–2004**	Increased female consumption leading to harmful effects in population	Middle-class lifestyles NHMRC guidelines are ‘risky’ Alcohol industry not as glamorous as it seems		
**2009–2011**	Regular consumption common—contributing to incidence increase Harmful effects need to be reported	Modern lifestyles Alcohol is the cause	Alcohol is to blame	Increase awareness
**2016–2018**	Consumption trends changing—for worse Serious consequences Alcohol industry disseminating misinformation	Entrenched in society Less familiar with link NHMRC guidelines not appropriate for all	Ignorant of link Alcohol industry finds fault with study claiming link	Increase awareness Moderation not abstinence Update/develop more suitable guidelines

Four shades are included in the scale (from light to dark): no prominence, mild prominence, moderate prominence, strong prominence.

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
