# Peer review of "How Are the Links between Alcohol Consumption and Breast Cancer Portrayed in Australian Newspapers?: A Paired Thematic and Framing Media Analysis"

_ijerph, 2021, doi:10.3390/ijerph18147657_

Round 1
Reviewer 1 Report
1. Introduction contains correctly defined research problem and justification of its actuality. The authors refer to up-to-date and valuable literature on the subject. At the end of this section, they formulate the objectives of their article. The aims of the article are clear and convincing.
2. Verses 94-95 report on the research method. I do not doubt that its application to public health can be regarded as innovative, nevertheless, I would like to point out that media content analysis in itself is a method that has long been known and used, especially in media studies. The same is true of Big Data technologies - for example, research by Adam Sadilek (University of Rochester), which used Big Data analysis of Twitter statuses to predict the onset and spread of the flu epidemic.
3) On sampling - I have no critical comments. In fact, my suggestions for improvements boil down to the comments in point 2. The article is well written, it can be published.
Author Response
We thank the reviewer for their review and positive comments on our paper. The reviewer only had one area for change, which related to the use of 'innovative' on lines 94-95. We agree with the reviewer that a traditional analysis of print or social media is not innovative - there are numerous papers undertaking various quantitative or content analyses of such data sources. However, our paper is innovative in its dual/paired analysis of print media by using both qualitative thematic analysis and framing analysis - that is not usual, certainly in public health research.
We have added an additional sentence in lines 95-96 to respond to the useful comment from the reviewer (the Tracked Changes version of the paper makes this easier to find):
"Whilst there are numerous content analyses of print and social media, our analysis is innovative due to its paired qualitative thematic and framing analysis"
Reviewer 2 Report
The subject of study seems to me more than interesting. In my opinion, studies related to public health and focused on improving the quality of life and well-being of citizens are very necessary, specifically in patients with breast cancer, a pathology with a very high incidence rate.
The title seems appropriate and meaningful to me, as well as attractive to readers. However, I do not believe that the print media are one of the main target communication media among the female population, as social networks can be today, for example: communication channels with the largest female audience in the age group studied.
About the introduction I have to say that I would change the structure. This part offers hypotheses without being identified as such or endorsed by experts in the field (lines 33 and 40, for example). In turn, assertions appear that should be documented (line 34: “historically”).
At line 45, I don't understand what kind of complexity the authors are referring to.
In my case, I would not absolutely rate alcohol as "best value" in reducing morbidity. It seems to me that, if it were correct, other specialists should endorse this statement.
I find that terms are mixed: media, written press, newspapers, magazines, ... In some paragraphs, indistinctly, one or the other is used, when, I understand that the article focuses on written press and not media, in general. In addition, I would make a difference between generalist newspapers and those specialized in health, since both their content and their audience are very diverse.
The subject is dealt with in a very generalized way, the audiences being very heterogeneous, due to their way of interacting with the media; the difference in age, social class, studies, ...; the stage of the disease; the type of media ...
I value the article for the complexity involved in a qualitative analysis and if the authors' objective is for health institutions, governments, the media or society to reflect on the issue. However, I believe that the research could be completed and strengthened with objective data on patient behavior, as well as with more current results, as it is a highly changing field. In addition, it would include private healthcare in the study.
Regarding materials and methodology, which table do the authors refer to on line 184? I would include it in the text. On the other hand, it would need further explanation on figure 2. It would offer a more complete methodological triangulation.
In the results, figure 3 causes me confusion: why do newspapers appear jointly analyzed? In the key, would categories be missing or would there be excess colors?
I would try to offer a greater depth in the results, as the study deserves it.

Author Response
We sincerely thanks the reviewer for their constructive and useful comments - they certainly improve the clarity of the paper. We repond to each point in turn, and we have added the changes as Track Changes to make it easier for the reviewer to see exactly how we have responded to their review.
- We agree that social media may also be useful for women in terms of getting information and that print newspapers are not the only source of information on alcohol or breast cancer. However, we decided to focus on newspaper articles because we felt that 45-64 year old women were more liekly to read newspapers. However, we recognise the potential limitation of only anlysing newspaper articles within the paper.
- We have deleted the word 'historically' from the paper
- We have deleted the word 'complex' from the paper
- We replaced the phrase 'best value' with 'good'
- We have dealt with our inconsistent use of words such as media and newspapers - when we are referring generally to things like media analysis and 'the media', we have kept the word 'media'. However, when referring to our particualr analysis, we have replaced 'media' with 'newspapers.
- The point about strengthening our analysis with additional data such as those on patient behaviour is a good one. However, we only had data from nespaper articles, so adding additional data is outside the scope of our paper.
- We have added a sentence on line 190 to exaplin the tables and link them to the tables in the Results section.
- Thankyou for picking up the problewm with Figure 3 - the formatting had changed and 2 of the categories in the legend were missing - this has now been rectified.
- A comment has been made about adding more depth to our analysis. We have re-read the paper and cannot see where additional deoth can occur whilst also keeping the paper as short as possible (it is already rather long because we present both qualitative thematic analysis and framing analysis).